

**Method to Quantify the Black Carbon Aerosol Light Absorption Enhancement with Entropy**
**and Diversity Measures**
Gang Zhao[1], Tianyi Tan[1], Yishu Zhu[1], Min Hu[1], Chunsheng Zhao[2*]
1 State Key Joint Laboratory of Environmental Simulation and Pollution Control, International Joint
Laboratory for Regional Pollution Control, Ministry of Education, College of Environmental
Sciences and Engineering, Peking University, Beijing, 100871, China
2 Department of Atmospheric and Oceanic Sciences, School of Physics, Peking University, Beijing,
100871, China
[*]**Correspondence author:** Chunsheng Zhao (zcs@pku.edu.cn)
**Abstract**
Large uncertainties remain when estimating the warming effects of ambient black carbon (BC)
aerosols on climate. One of the key challenges in modeling the radiative effects is predicting the BC
light absorption enhancement, which is mainly determined by its mass ratio of non-BC coating
thickness to BC (MR). For the same MR, recent researches find that the radiative absorption
enhancements by BC are also controlled by its particle-to-particle heterogeneity. In this study, the
BC mixing state index ($\chi$) is developed to quantify the dispersion of ambient black carbon aerosol
mixing states based on binary systems of BC and other non-black carbon components. We
demonstrate that the BC light absorption enhancement increases with $\chi$ for the same MR, which
indicates that $\chi$ can be employed as a factor to constrain the light absorption enhancement of ambient
BC. Our framework can be further used in the model to study the black carbon radiative effects on
climate change.
**1 Introduction**
Black carbon (BC) aerosols absorb solar radiation, thus exert warming effects on the earth's
energy system (Bond and Bergstrom, 2006;Bond et al., 2013). However, large uncertainties remain
when quantifying the BC warming effects (Cui et al., 2016;Jacobson, 2010;Koch et al., 2009;Menon



et al., 2002). Most of the BC particles were emitted from incomplete combustion of bio fossil fuel
(Bond et al., 2013). After initially emitted, the BC particles would experience aging processing with
some other non-BC components coated on the BC particles (Peng et al., 2017;Peng et al., 2016).
During the aging processing, the light absorption of BC aerosols would increase, which is well
known as "lensing effects" (Saleh et al., 2013;Saleh et al., 2014). One critical challenge in estimating
the BC warming effects is quantifying the "lensing effects" of ambient BC aerosols (Liu et al., 2017).

The light absorption enhancement ($E_{abs}$), which is the ratio of light absorption of BC aerosols

with the coating to that of bare BC particles, is proposed to quantify the "lensing effects".
Comprehensive studies have been carried out to study the $E_{abs}$ (Liu et al., 2017;Peng et al.,
2016;Liu et al., 2015;Fierce et al., 2016;Fierce et al., 2020;Cappa et al., 2012). However, a large
discrepancy remains between the results of $E_{abs}$ from field measurements and laboratory studies.
The measured $E_{abs}$ of laboratory generated monodisperse BC particles can reach up to a factor of 2,
which is consistent with the results from the Mie scattering model (Cappa et al., 2012;Cappa et al.,
2019). However, some field measurement shows that the $E_{abs}$ of ambient BC aerosols are relatively
small, with 1.06 at California (Cappa et al., 2012), 1.07 in South China (Lan et al., 2013), and 1.10 in
Japan (Nakayama et al., 2014), while the measured $E_{abs}$ of ambient BC reaches 1.59 during
summer time in Beijing (Xie et al., 2019).
Many factors, such as the morphology of the BC core, the position of BC core inside coating, the
coating thickness, and size distribution of the BC, would influence the $E_{abs}$ of ambient BC aerosols.
Wu et al. (2018) reported that the BC light absorption properties vary significantly for different
morphology from the calculation of models. Laboratory studies also find that the light absorption
properties of the BC core were tuned due to the change of the BC core morphology (Yuan et al.,
2020). Comparing with the concentric spherical structure, the off-center coated BC aggregates would
lead to up to a 31% reduction in $E_{abs}$ by the multiple-sphere T-matrix method (Zhang et al., 2017).
It has been well studied that the $E_{abs}$ is highly related with the mass ratio of coating materials and
BC core (MR) (Liu et al., 2014;Liu et al., 2017). Zhao et al. (2019b) reported that the light


absorption properties of ambient BC particles are influenced by BC mass size distribution. Besides,
recently researchers found that the $E_{abs}$ are also controlled by particle-to-particle heterogeneity
(Fierce et al., 2016;Fierce et al., 2020). As shown in Fig.1, the $E_{abs}$ of ambient aerosols for the
same MR would vary by about 30%, which is consistency with the results of Fierce et al. (2020).
However, there is no study, to our best knowledge, that constrains the uncertainties of the $E_{abs}$ for
the same MR.

In this study, we developed a BC mixing states index ($\chi$) to quantify the dispersion of black

carbon aerosol mixing states based on binary systems of BC and other non-black carbon components.
We demonstrate that the BC $E_{abs}$ increases with $\chi$ for the same MR based on the field measurement,
which indicates that $\chi$ can be employed as a factor to constrain the $E_{abs}$ properties of ambient BC.
**2 Data and methods**
**2.1 Field measurement**

The field measurements were conducted at a suburban site Taizhou (119°57' E, 32°35' N) from

26 May to 18 June. As shown in Fig. S1, the Taizhou site lies between two large cities of Nanjing
and Shanghai, where the aerosols can be seen as representative that of the Yangtze River Delta area
(Liu et al., 2020). More details of the field measurements can refer to Zhao et al. (2019a). During the
field measurement, we placed all of the instruments in a container where the temperature was
carefully controlled between 22 and 26 ℃. A PM$_{10}$ impactor, which is about 5 meters above the
ground, was mounted on the top of the container. The sample aerosols were drawn from the impactor
and then dried by a Nafion dryer tube.

The size-resolved BC mixing states were measured by using a differential mobility analyzer

(DMA, model 3081, TSI, USA) in tandem with a single-particle soot photometer (SP2, Droplet
Measurement Technologies, USA). Detailed information on the DMA can refer to Zhao et al.
(2019c). SP2 can measure the BC mass concentration from the incandescence signals emitted by the
BC particle, which is heated to around 6000 K by laser with a wavelength of 1064 nm (Zhao et al.,
2020b). Along with the measurement of size-resolved BC mixing states, a nephelometer (Aurora 300,



Ecotech, Australia) (Müller et al., 2011) was employed to measure the aerosol scattering coefficient
($\sigma_{sca}$) at the wavelength of 525 nm.
**2.2 BC mixing states from DMA-SP2 system**

In this study, the SP2 was placed after the DMA to measure the size-selected mixing states of the

quasi-monodisperse aerosols. The schematic instrument setup is shown in Fig. S1 and the details can
refer to part 1 in the supplementary material. After careful calibrations of the SP2 (part 2.1 in the
supplementary material), transformations of the measured signals to BC mass concentrations (part
2.2 in the supplementary material), and multiple charging corrections (part 2.3 in the supplementary
material), the BC-containing number concentration distribution under different total diameter ($Dp$)
and BC core diameter ($Dc$) can be calculated, as shown in Fig. S4 (b). The details of the calculation
of size-resolved BC mixing states from the DMA-SP2 system can refer to Zhao et al. (2020a). The
measured size-resolved BC mixing states as in Fig. S4(b) were used for further analysis. It should be
mentioned that the measured number distribution of BC-containing aerosols is two dimensional
($\frac{dN}{dlogDp \cdot dlogDc}$).
**2.3 Calculating the aerosol optical properties**
**2.3.1 Calculating the aerosol absorption coefficient for a given $Dp$ and $Dc$**

A Mie scattering model (Bohren and Huffman, 2007) was employed to calculate the aerosol

absorption coefficient ($\sigma_{abs}$). When calculating the $\sigma_{abs}$ of single particle, the Mie scattering model
requires the diameter of the core, the coating thickness, the refractive index of the core, and the
refractive index of the shell. The refractive index of the core adopted here is $1.67+0.67i$, which is the
calculated mean value by comparing the measured light absorption and calculated light absorption
properties (Zhao et al., 2020a). The refractive index of the shell is chosen to be $1.46+0i$, which is
assumed to be as that of the non-BC component measured by the DMA-SP2 system (Zhao et al.,
2019a;Zhao et al., 2019c). With the above information, the $\sigma_{abs}$ values at a given $Dp$ and a given $Dc$
can be calculated.



### 2.3.2 Calculating the aerosol bulk absorption coefficient

We calculate the single-particle $\sigma_{abs}$ of different Dp and Dc with the given refractive index of core and shell and then the ambient aerosol $\sigma_{abs}$ distributions at different Dp and Dc $\left(\frac{d\sigma_{abs}}{dlogDp \cdot dlogDc}\right)$ can be calculated by multiplying the number concentrations of the BC-contained aerosols $\left(\frac{dN}{dlogDp \cdot dlogDc}\right)$. By integrating the $\frac{d\sigma_{abs}}{dlogDp \cdot dlogDc}$ over different Dc values, the ambient aerosol $\sigma_{abs}$ distribution along with different Dp $\left(\frac{d\sigma_{abs}}{dlogDp}\right)$ can be calculated. The total $\sigma_{abs}$ of the ambient BC-containing aerosols can be calculated by integrating the $\frac{d\sigma_{abs}}{dlogDp}$ over different Dp values.

### 2.3.3 Calculating the aerosol $E_{abs}$

Along with calculating the $\sigma_{abs,Dp,Dc}$ of single-particle for different Dp and Dc, we calculate the corresponding light absorption ($\sigma_{abs, Dc, Dc}$) value for Dc without thickness. The corresponding total light absorption of all measured BC-contained aerosols without thickness can be calculated by integrating the calculated $\sigma_{abs, Dc, Dc}$ among different Dp and Dc weighted with $\frac{dN}{dlogDp \cdot dlogDc}$. Thus the ambient BC particles without coating ($\sigma_{abs,Dp=0}$) can be calculated. The bulk ambient aerosol $E_{abs}$ can thus be calculated with $E_{abs} = \frac{\sigma_{abs}}{\sigma_{abs, Dp=0}}$.

### 2.4 Quantifying the dispersion of BC mixing states

As for BC particles with known Dp and Dc, the mass concentration of BC core and coating material can be calculated with the effective density of BC core and coating material. The effective density of the BC core is calculated in detail in section 2.2 in the supplement. The effective density of the coating material is assumed to be the same as the measured effective density of non-BC aerosols by using a centrifugal particle mass analyzer (version 1.53, Cambustion Ltd, UK) in tandem with a scanning mobility particle sizer system (Zhao et al., 2019a) and a mean value of 1.5 g/cm³ was used here.



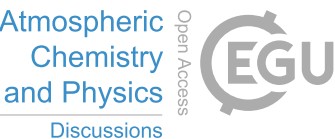

For each of the particle $i$ $(i=1,2,..,$ $N$ is the measured BC-containing aerosol number
concentration), we can calculate its mass ratio of BC with
$$p_{i,BC} = \frac{m_{i,BC}}{m_i},$$    (1)
where $m_{i,BC}$ is the mass concentration of BC and $m_i$ is the total mass concentration of particle $i$.
The mass portion of BC can be calculated as
$$p_{BC} = \frac{m_{BC}}{m_{tot}},$$    (2)
were $m_{BC}$ (the total mass concentration of BC) and $m_{tot}$ (total mass of BC-containing aerosols)
can be calculated as $m_{BC} = \sum_{i=1}^{N} m_{i,BC}$, $m_{tot} = \sum_{i=1}^{N} m_i$. The MR is calculated as:
$$MR = \frac{(m_{tot} - m_{BC})}{m_{BC}},$$    (3)
The mass portion of particle $i$ to total BC-containing aerosols is calculated as
$$p_i = \frac{m_i}{m_{tot}}.$$    (4)
With the definition above, we can calculate the mixing entropy of particle $i$ ($H_i$) by:
$$H_i = - (p_{i,BC} ln(p_{i,BC}) + (1 - p_{i,BC}) ln(1 - p_{i,BC}),$$    (5)
the average mixing entropy of each particle by:
$$H_\alpha = \sum_{i=1}^{N} p_i H_i,$$    (6)
And the population bulk mixing entropy by:
$$H_\gamma = - (p_{BC} ln(p_{BC}) + (1 - p_{BC}) ln(1 - p_{BC}).$$    (7)
Then the average particle species diversity can be calculated by
$$D_\alpha = e^{H_\alpha},$$    (8)
And the bulk population species diversity can be calculated by
$$D_\gamma = e^{H_\gamma},$$    (9)
With the above information, the dispersion of BC particle mixing states can be defined as
$$\chi = \frac{D_\alpha - 1}{D_\gamma - 1}.$$    (10)



The basic idea of quantifying the BC particle mixing states is the same as that of Riemer and
West (2013) and Riemer et al. (2019), their framework mainly focuses on the bulk ambient aerosols
with about five species (Bondy et al., 2018;Ye et al., 2018). Our developed χ is a reduced parameter
that only concerns the BC-containing aerosols with two species of BC component and non-BC
coating materials.
**3. Results and Discussions**
**3.1 BC mixing states diagram**
A mixing state diagram as shown in Fig. 2 was employed for better understanding the dispersion
of BC mixing states. Nine different group bulks of aerosols were given and summarized in Table 1.
For each group, we include six BC-containing particles with different mass concentrations of BC
core and non-BC coating material.
For group 1, the amounts of BC are very small (near zero) and most of the aerosols are
composed of the non-BC component. The $D_\alpha$ and $D_\gamma$ values are 1.00 and 1.00 respectively. These
groups can also be described as all of the particles are pure BC particles without coating.
For groups 2, 3, and 4, the mass concentration ratios of the BC component to the non-BC
component are 1:5, 2:4, and 3:3 respectively. All of the $D_\alpha$ values are 1.00 for groups 2, 3, and 4
because the BC particles are externally mixed. The corresponding $D_\gamma$ values are 1.56, 1.89, and
2.00 respectively. For these three groups, the χ values are all 0.00.
For groups 4, 5, 6, and 7, the mass concentration ratios of the BC component to the non-BC
component are all 1:1 while the BC component is mixed to a different extent. It is easy to conclude
that the BC particles of group 7 are most well mixed among these four groups. The corresponding χ
values are 0, 0.26, 0.83, and 1.0 for group 4, 5, 6, and 7, respectively.
As for groups 8 and 9, the mass concentration ratios of the BC component to the non-BC
component are 1:6.1. The $D_\gamma$ values are 1.5 and the $D_\alpha$ values are 1.5 and 1.35 respectively.
From the different group, the average particle species diversity $D_\gamma$ value is mainly determined
by the total mass concentration ratio of the BC component to the non-BC component. It varies


between 1 and 2 for different total mass concentration ratios. The $D_\gamma$ increases when the mass ratio
approaches 1. The bulk population species diversity $D_\alpha$ ranges between 1 and $D_\gamma$. It denotes the
diversity of different BC-containing particles.

**3.2 Overview of the measurement**

Fig.S6 gives the time series of our field measurements results. During the field measurement, the
$\sigma_{sca}$ varies between 29 and 1590 Mm$^{-1}$. The ranges of $H_\alpha$, $H_\gamma$, $D_\alpha$, $D_\gamma$, and χ are 0.10~0.55,
0.42~0.64, 1.32~1.72, 1.52~1.91 and 0.62~0.82 respectively.
For a better understanding of the characteristics of the above parameters, we only present the
time series of these parameters during a pollution period between 27, May and 30, May in Fig. 3. As
shown in Fig. 3, the MR increased from about 2 to 4 when the $\sigma_{sca}$ increased from 300 to 1200
Mm$^{-1}$, which indicates that some secondary aerosol components were coated on the BC particles
when the ambient air is more polluted. During the aging processing, the $H_\alpha$ decreased from 0.51 to
0.38 and $H_\gamma$ decreased from 0.63 to 0.49. The $D_\alpha$ decreases with the MR from 1.66 to 1.48, which
is consistent with the results in section 3.1 that the $D_\alpha$ should decrease with the MR when the MR is
larger than 1. The χ varies between 0.68 and 0.79. It is worth noting that the χ is not well correlated
with the pollution conditions.
The daily variation of $\sigma_{sca}$, which is highly related to the development of the boundary layer,
reaches its maximum value of 525 Mm$^{-1}$ at 6:00 AM and a minimum value of 150 Mm at 7:00 PM.
The daily variation of MR is largest at 5:00 AM with a mean value of 3.16 and reaches its minimum
value of 2.56 at 7:00 PM. The daily variation of MR was mainly influenced by aging processing and
anthropogenic activities. During the daytime, the newly emitted BC particles due to anthropogenic
activities have low MR and the measured mean MR is low than that at night. The $D_\alpha$ values, which
are anti-correlated with MR, show the opposite trend with MR. As for χ, it is smaller in the daytime
than that at night. The lower χ values at daytime mainly resulted from the mixing of newly emitted
BC particles due to anthropogenic activities and some pre-existed aged BC particles.


### 3.3 Relationship between the $\chi$ and $E_{abs}$ from measurement

For each of the measured group of size-resolved BC mixing states, we calculated the corresponding MR, $\chi$, and $E_{abs}$. And the relationship between the MR and absorption enhancement is summarized in Fig. 5. Overall, the BC $E_{abs}$ increase with MR, which is consistent with the previous knowledge. For a given value of MR, $E_{abs}$ varies by about 20%, especially for these conditions with MR larger than 1.0. When MR is larger than 1.0, the $E_{abs}$ increase with the $\chi$. Relationship between the $E_{abs}$ and $\chi$ is rather complex when MR is smaller than 1.0. However, only 448 of 6948 groups (6.4%) of the measured MR values are smaller than 1. Therefore, for most of the conditions, the measured $E_{abs}$ should increase with $\chi$, which indicates that the refractive index of $\chi$ can be employed as a factor to constrain the $E_{abs}$ of ambient aerosols.

A schematic diagram as shown in Fig. 6 to denotes the relationship between the $E_{abs}$ and $\chi$. From Fig. 6, we calculated the $E_{abs}$ and $\chi$ under differ MR and then compared the $E_{abs}$ of different bulk aerosols. The first group contains two particles with both the MR equaling 8. The corresponding $\chi$ is 1.00 and $E_{abs}$ is 1.60. Another group of particles contains two particles with MR equaling 1 and 15, respectively. Thus the second group of particles has a mean MR of 8. The calculated corresponding $\chi$ and $E_{abs}$ are 0.79 and 1.42 respectively. Thus, the $E_{abs}$ tend to increase with $\chi$ for the same MR, which is mainly resulted from that the increasing ratio of $E_{abs}$ (the slope of $E_{abs}$ to MR) decrease with MR.

It is worth noting that the increasing ratio is almost the same when the MR is in the range of 0 and 3. Therefore, the $E_{abs}$ doesn't tend to increase with the $\chi$ when the MR was less than 1, which is consistent with our study as shown in Fig. 6.

### 3.4 Relationship between the $\chi$ and $E_{abs}$ from simulation

A Mont-Carlo simulation was carried out for a better understanding of the relationship between $\chi$ and $E_{abs}$. During the simulation, the number of BC-containing particles was assumed to be 30. For each of the BC particle, the core diameter of the BC particle was randomly generated with a geometric mean diameter of 130.7 nm and a geometric standard deviation of 1.5, which is the mean





measurement results of the BC core distribution during the field measurement (Zhao et al., 2020b).
The corresponding MR of the BC particle is assumed to be in the range between 0.0 (pure BC
particles without coating) and 78.0 (particles with a core diameter of 130 nm and a total diameter of
560 nm). For each of the group of particles, the corresponding aerosol bulk MR, $E_{abs}$ and χ can be
calcualted. The simulations were conducted for $10^7$ times, and the calculated mean and standard
deviation of $E_{abs}$ under different MR and χ are summarized in Fig. 7 (a) and (b).

From Fig. 7 (a), the calculated $E_{abs}$ tend to increase with MR for each of the given χ, which is

consistent with the previous knowledge of the BC light absorption properties. Then the MR is
smaller than 2, the calculated $E_{abs}$ does not seem to increase with the χ, which is consistent with the
analyzed results from section 3.3 and Fig. 6. When the MR is larger 2, the $E_{abs}$ tend to increase
with the χ. The larger the MR is, the $E_{abs}$ is more sensitive to χ. Two reasons may lead to this
phenomenon. One reason is that that calculated slope of $E_{abs}$ to MR for one particle as shown in Fig.
6 decreases with the MR. Another reason is that the calculated $E_{abs}$ range increase with MR when
the χ changes between 0 and 1 as shown in Fig. 5.

As for the uncertainties of simulated $E_{abs}$, it tends to increase with the MR, which is consistent

with the previous discussions that the $E_{abs}$ the range tends to increase with MR. Overall, the
calculated standard deviations of $E_{abs}$ are all the way smaller than 10% for different MR and χ.
Therefore, the calculated $E_{abs}$ can be well constrained by χ.

**4 Conclusion**

Larger uncertainties remain when estimating the warming effects of ambient BC aerosols due to

the poor understanding of the ambient BC light absorption enhance ratio. Previous studies find that
the light absorption of ambient aerosols was mainly determined by the morphology of the BC core,
the position of the BC core inside coating, the coating thickness, and the size distribution of the BC.
We find that there are more than 20% of uncertainties for the same measured mean coating thickness,
i.e. the same measured MR based on the field measurement of the size-resolved BC mixing states.
However, there were no-study, to our best knowledge, that attempts to constrain the uncertainties.



In this study, we developed the BC mixing states index χ based on the mass concentrations of
BC components and non-BC material of each BC-containing particle. Results show that the light
absorption enhancement ratio $E_{abs}$  tend to increase the χ for the same measured MR. Therefore, our
developed parameter χ, which reflects the dispersion of the BC mixing states, can be employed as an
effective parameter to constrain the light absorption enhancement of ambient BC-containing
aerosols.

*Data availability.* The data involved is available in the manuscript.
*Author contributions.* Gang Zhao wrote the manuscript. Chunsheng Zhao, Min Hu, Tianyi Tan,
Song Guo, Zhijun Wu, Yishu Zhu and Gang Zhao discussed the results.
*Competing interests.* The authors declare that they have no conflict of interest.
*Acknowledgments.* This work is supported by the National Key R&D Program of China
(2016YFC020000: Task 5) and the National Natural Science Foundation of China (41590872).



269 Bohren, C. F., and Huffman, D. R.: Absorption and Scattering by a Sphere, in Absorption and

270 Scattering of Light by Small Particles, Wiley-VCH Verlag GmbH, 82-129, 2007.

271 Bond, T. C., and Bergstrom, R. W.: Light Absorption by Carbonaceous Particles: An Investigative

272 Review, Aerosol Sci. Technol., 40, 27-67, 10.1080/02786820500421521, 2006.

273 Bond, T. C., Doherty, S. J., Fahey, D. W., Forster, P. M., Berntsen, T., DeAngelo, B. J., Flanner, M.

274 G., Ghan, S., Karcher, B., Koch, D., Kinne, S., Kondo, Y., Quinn, P. K., Sarofim, M. C., Schultz, M.

275 G., Schulz, M., Venkataraman, C., Zhang, H., Zhang, S., Bellouin, N., Guttikunda, S. K., Hopke, P.

276 K., Jacobson, M. Z., Kaiser, J. W., Klimont, Z., Lohmann, U., Schwarz, J. P., Shindell, D.,

277 Storelvmo, T., Warren, S. G., and Zender, C. S.: Bounding the role of black carbon in the climate

278 system: A scientific assessment, J Geophys Res-Atmos, 118, 5380-5552, 10.1002/jgrd.50171, 2013.

279 Bondy, A. L., Bonanno, D., Moffet, R. C., Wang, B., Laskin, A., and Ault, A. P.: The diverse

280 chemical mixing state of aerosol particles in the southeastern United States, Atmospheric Chemistry

281 and Physics, 18, 12595-12612, 10.5194/acp-18-12595-2018, 2018.

282 Cappa, C. D., Onasch, T. B., Massoli, P., Worsnop, D. R., Bates, T. S., Cross, E. S., Davidovits, P.,

283 Hakala, J., Hayden, K. L., Jobson, B. T., Kolesar, K. R., Lack, D. A., Lerner, B. M., Li, S. M.,

284 Mellon, D., Nuaaman, I., Olfert, J. S., Petaja, T., Quinn, P. K., Song, C., Subramanian, R., Williams,

285 E. J., and Zaveri, R. A.: Radiative Absorption Enhancements Due to the Mixing State of

286 Atmospheric Black Carbon, Science, 337, 1078-1081, 10.1126/science.1223447, 2012.

287 Cappa, C. D., Zhang, X., Russell, L. M., Collier, S., Lee, A. K. Y., Chen, C.-L., Betha, R., Chen, S.,

288 Liu, J., Price, D. J., Sanchez, K. J., McMeeking, G. R., Williams, L. R., Onasch, T. B., Worsnop, D.

289 R., Abbatt, J., and Zhang, Q.: Light Absorption by Ambient Black and Brown Carbon and its

290 Dependence on Black Carbon Coating State for Two California, USA, Cities in Winter and Summer,

291 Journal of Geophysical Research: Atmospheres, 124, 1550-1577, 10.1029/2018jd029501, 2019.

292 Cui, X., Wang, X., Yang, L., Chen, B., Chen, J., Andersson, A., and Gustafsson, Ö.: Radiative

293 absorption enhancement from coatings on black carbon aerosols, Science of The Total Environment,

294 551-552, 51-56, doi.org/10.1016/j.scitotenv.2016.02.026, 2016.





Fierce, L., Bond, T. C., Bauer, S. E., Mena, F., and Riemer, N.: Black carbon absorption at the global
scale is affected by particle-scale diversity in composition, Nature communications, 7, 12361,
10.1038/ncomms12361, 2016.
Fierce, L., Onasch, T. B., Cappa, C. D., Mazzoleni, C., China, S., Bhandari, J., Davidovits, P.,
Fischer, D. A., Helgestad, T., Lambe, A. T., Sedlacek, A. J., 3rd, Smith, G. D., and Wolff, L.:
Radiative absorption enhancements by black carbon controlled by particle-to-particle heterogeneity
in composition, Proceedings of the National Academy of Sciences of the United States of America,
117, 5196-5203, 10.1073/pnas.1919723117, 2020.
Jacobson, M. Z.: Short-term effects of controlling fossil-fuel soot, biofuel soot and gases, and
methane on climate, Arctic ice, and air pollution health, Journal of Geophysical Research:
Atmospheres, 115, n/a-n/a, 10.1029/2009JD013795, 2010.
Koch, D., Schulz, M., Kinne, S., and Mcnaughton, C.: Evaluation of black carbon estimations in
global aerosol models, Atmospheric Chemistry & Physics, 9, 9001-9026, 2009.
Lan, Z.-J., Huang, X.-F., Yu, K.-Y., Sun, T.-L., Zeng, L.-W., and Hu, M.: Light absorption of black
carbon aerosol and its enhancement by mixing state in an urban atmosphere in South China,
Atmospheric Environment, 69, 118-123, 10.1016/j.atmosenv.2012.12.009, 2013.
Liu, D., Allan, J. D., Young, D. E., Coe, H., Beddows, D., Fleming, Z. L., Flynn, M. J., Gallagher, M.
W., Harrison, R. M., Lee, J., Prevot, A. S. H., Taylor, J. W., Yin, J., Williams, P. I., and Zotter, P.:
Size distribution, mixing state and source apportionment of black carbon aerosol in London during
wintertime, Atmospheric Chemistry and Physics, 14, 10061-10084, 10.5194/acp-14-10061-2014,

315   2014.

Liu, D., Whitehead, J., Alfarra, M. R., Reyes-Villegas, E., Spracklen, Dominick V., Reddington,
Carly L., Kong, S., Williams, Paul I., Ting, Y.-C., Haslett, S., Taylor, Jonathan W., Flynn, Michael J.,
Morgan, William T., McFiggans, G., Coe, H., and Allan, James D.: Black-carbon absorption
enhancement in the atmosphere determined by particle mixing state, Nature Geoscience, 10, 184-188,
10.1038/ngeo2901, 2017.



Liu, J., Li, X., Li, D., Xu, R., Gao, Y., Chen, S., Liu, Y., Zhao, G., Wang, H., Wang, H., Lou, S.,
Chen, M., Hu, J., Lu, K., Wu, Z., Hu, M., Zeng, L., and Zhang, Y.: Observations of glyoxal and
methylglyoxal in a suburban area of the Yangtze River Delta, China, Atmospheric Environment, 238,
117727, 10.1016/j.atmosenv.2020.117727, 2020.
Liu, S., Aiken, A. C., Gorkowski, K., Dubey, M. K., Cappa, C. D., Williams, L. R., Herndon, S. C.,
Massoli, P., Fortner, E. C., Chhabra, P. S., Brooks, W. A., Onasch, T. B., Jayne, J. T., Worsnop, D.
R., China, S., Sharma, N., Mazzoleni, C., Xu, L., Ng, N. L., Liu, D., Allan, J. D., Lee, J. D., Fleming,
Z. L., Mohr, C., Zotter, P., Szidat, S., and Prevot, A. S.: Enhanced light absorption by mixed source
black and brown carbon particles in UK winter, Nature communications, 6, 8435,
10.1038/ncomms9435, 2015.
Menon, S., Hansen, J., Nazarenko, L., and Luo, Y.: Climate effects of black carbon aerosols in China
and India, Science, 297, 2250-2253, 10.1126/science.1075159, 2002.
Müller, T., Laborde, M., Kassell, G., and Wiedensohler, A.: Design and performance of a
three-wavelength LED-based total scatter and backscatter integrating nephelometer, Atmos. Meas.
Tech., 4, 1291-1303, 10.5194/amt-4-1291-2011, 2011.
Nakayama, T., Ikeda, Y., Sawada, Y., Setoguchi, Y., Ogawa, S., Kawana, K., Mochida, M., Ikemori,
F., Matsumoto, K., and Matsumi, Y.: Properties of light-absorbing aerosols in the Nagoya urban area,
Japan, in August 2011 and January 2012: Contributions of brown carbon and lensing effect, Journal
of Geophysical Research: Atmospheres, 119, 12,721-712,739, 10.1002/2014JD021744, 2014.
Peng, J., Hu, M., Guo, S., Du, Z., Zheng, J., Shang, D., Levy Zamora, M., Zeng, L., Shao, M., Wu,
Y.-S., Zheng, J., Wang, Y., Glen, C. R., Collins, D. R., Molina, M. J., and Zhang, R.: Markedly
enhanced absorption and direct radiative forcing of black carbon under polluted urban environments,
Proceedings of the National Academy of Sciences, 113, 4266-4271, 10.1073/pnas.1602310113,

344  2016.

Peng, J., Hu, M., Guo, S., Du, Z., Shang, D., Zheng, J., Zheng, J., Zeng, L., Shao, M., Wu, Y.,
Collins, D., and Zhang, R.: Ageing and hygroscopicity variation of black carbon particles in Beijing





measured by a quasi-atmospheric aerosol evolution study (QUALITY) chamber, Atmospheric
Chemistry and Physics, 17, 10333-10348, 10.5194/acp-17-10333-2017, 2017.
Riemer, N., and West, M.: Quantifying aerosol mixing state with entropy and diversity measures,
Atmospheric Chemistry and Physics, 13, 11423-11439, 10.5194/acp-13-11423-2013, 2013.
Riemer, N., Ault, A. P., West, M., Craig, R. L., and Curtis, J. H.: Aerosol Mixing State:
Measurements, Modeling, and Impacts, Reviews of Geophysics, 57, 187-249,
10.1029/2018rg000615, 2019.
Saleh, R., Hennigan, C. J., McMeeking, G. R., Chuang, W. K., Robinson, E. S., Coe, H., Donahue, N.
M., and Robinson, A. L.: Absorptivity of brown carbon in fresh and photo-chemically aged
biomass-burning emissions, Atmos. Chem. Phys., 13, 7683-7693, 10.5194/acp-13-7683-2013, 2013.
Saleh, R., Robinson, E. S., Tkacik, D. S., Ahern, A. T., Liu, S., Aiken, A. C., Sullivan, R. C., Presto,
A. A., Dubey, M. K., Yokelson, R. J., Donahue, N. M., and Robinson, A. L.: Brownness of organics
in aerosols from biomass burning linked to their black carbon content, Nature Geoscience, 7, 647,
10.1038/ngeo2220, 2014.
Wu, Y., Cheng, T., Liu, D., Allan, J. D., Zheng, L., and Chen, H.: Light Absorption Enhancement of
Black Carbon Aerosol Constrained by Particle Morphology, Environ Sci Technol, 52, 6912-6919,
10.1021/acs.est.8b00636, 2018.
Xie, C., Xu, W., Wang, J., Liu, D., Ge, X., Zhang, Q., Wang, Q., Du, W., Zhao, J., Zhou, W., Li, J.,
Fu, P., Wang, Z., Worsnop, D., and Sun, Y.: Light absorption enhancement of black carbon in urban
Beijing in summer, Atmospheric Environment, 10.1016/j.atmosenv.2019.06.041, 2019.
Ye, Q., Gu, P., Li, H. Z., Robinson, E. S., Lipsky, E., Kaltsonoudis, C., Lee, A. K. Y., Apte, J. S.,
Robinson, A. L., Sullivan, R. C., Presto, A. A., and Donahue, N. M.: Spatial Variability of Sources
and Mixing State of Atmospheric Particles in a Metropolitan Area, Environ Sci Technol, 52,
6807-6815, 10.1021/acs.est.8b01011, 2018.



Yuan, C., Zheng, J., Ma, Y., Jiang, Y., Li, Y., and Wang, Z.: Significant restructuring and light
absorption enhancement of black carbon particles by ammonium nitrate coating, Environ Pollut, 262,
114172, 10.1016/j.envpol.2020.114172, 2020.
Zhang, X., Mao, M., Yin, Y., and Wang, B.: Absorption enhancement of aged black carbon aerosols
affected by their microphysics: a numerical investigation, Journal of Quantitative Spectroscopy and
Radiative Transfer, 202, 90-97, 10.1016/j.jqsrt.2017.07.025, 2017.
Zhao, G., Tan, T., Zhao, W., Guo, S., Tian, P., and Zhao, C.: A new parameterization scheme for the
real part of the ambient urban aerosol refractive index, Atmos. Chem. Phys., 19, 12875-12885,
10.5194/acp-19-12875-2019, 2019a.
Zhao, G., Tao, J., Kuang, Y., Shen, C., Yu, Y., and Zhao, C.: Role of black carbon mass size
distribution in the direct aerosol radiative forcing, Atmos. Chem. Phys., 19, 13175-13188,
10.5194/acp-19-13175-2019, 2019b.
Zhao, G., Zhao, W., and Zhao, C.: Method to measure the size-resolved real part of aerosol refractive
index using differential mobility analyzer in tandem with single-particle soot photometer,
Atmospheric Measurement Techniques, 12, 3541-3550, 10.5194/amt-12-3541-2019, 2019c.
Zhao, G., Li, F., and Zhao, C.: Determination of the refractive index of ambient aerosols,
Atmospheric Environment, 240, 117800, 10.1016/j.atmosenv.2020.117800, 2020a.
Zhao, G., Shen, C., and Zhao, C.: Technical note: Mismeasurement of the core-shell structure of
black carbon-containing ambient aerosols by SP2 measurements, Atmospheric Environment, 243,
117885, 10.1016/j.atmosenv.2020.117885, 2020b.


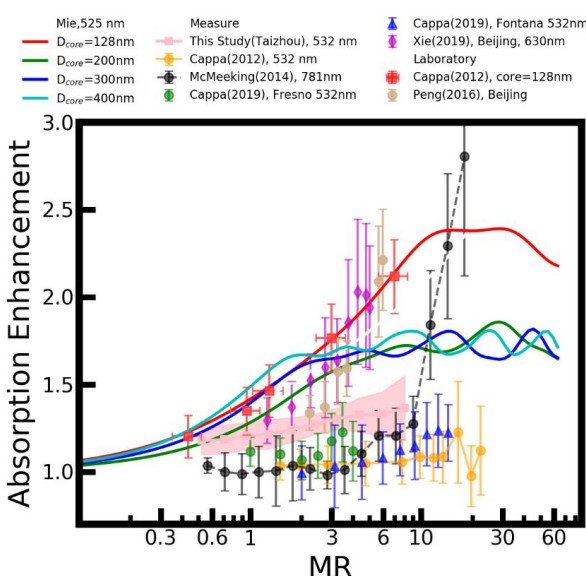


Figure 1. The measured $E_{abs}$ of BC particles from different ambient measurements, including this

work (in pink), and lab studies.


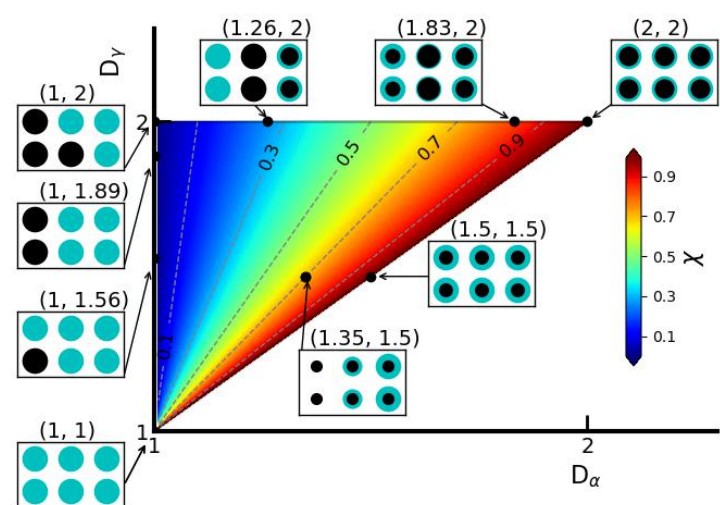


**Figure 2.** Mixing states diagram to illustrate the relationship between $D_\alpha$, $D_\gamma$, and χ. Each species

consists of six particles, and the colors of black and cyan represent the BC and non-BC components.




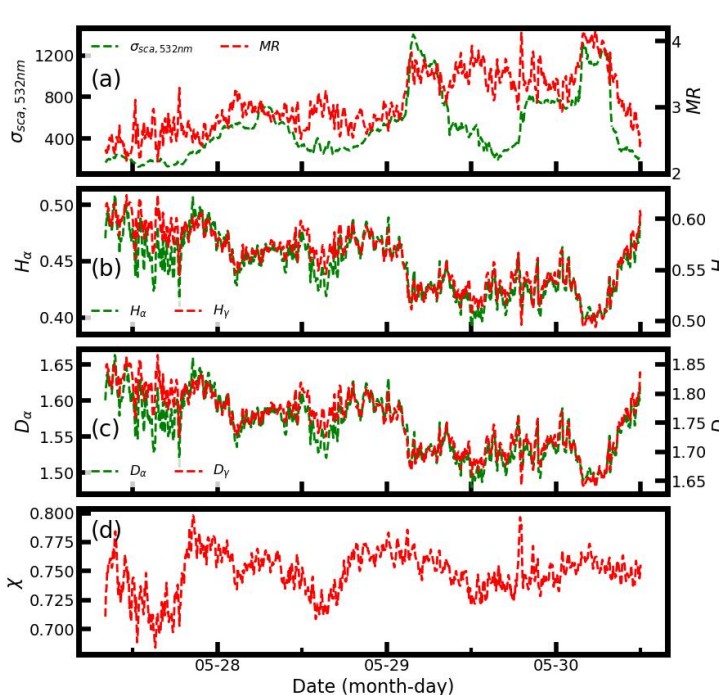


**Figure 3.** Measured time series of (a) $\sigma_{sca}$ and MR, (b) $H_\alpha$ and $H_\gamma$, (c) $D_\alpha$ and $D_\gamma$, and (d) $\chi$.


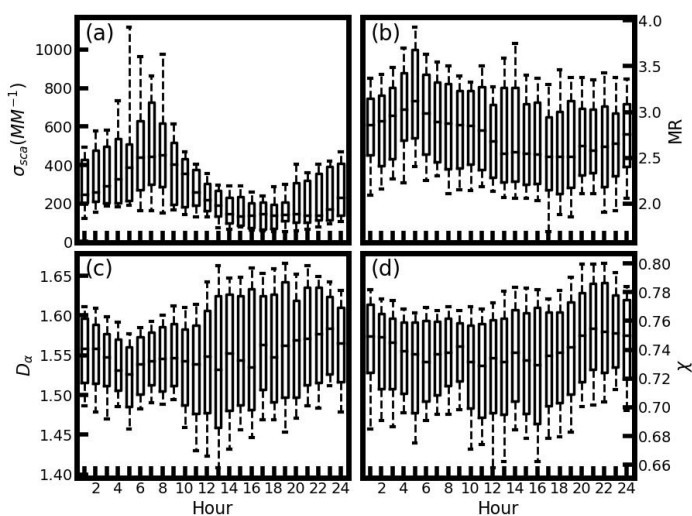


**Figure 4.** Daily variation of the measured (a) $\sigma_{sca}$, (b) MR, (c) $D_{\alpha}$, and (d) χ.




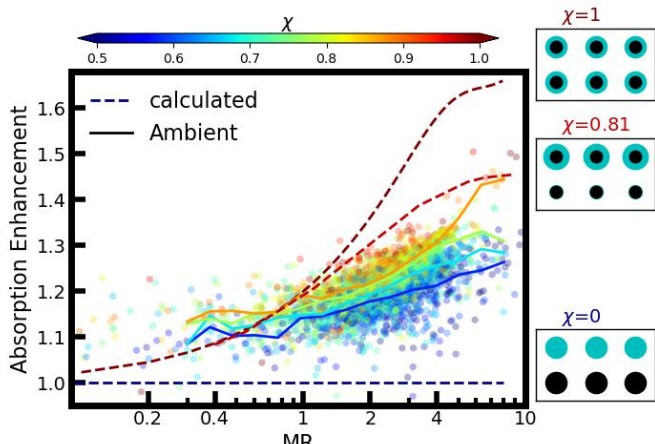


**Figure 5.** Relationship between the BC $E_{abs}$ and the measured mass ratio of the BC-containing
aerosols coating material to BC under different $\chi$ conditions. Four solid lines from bottom to up
corresponding to the measured ambient size-resolved BC mixing states data with $\chi$ ranges of
0.575~0.625, 0.625~0.675, 0.675~0.725, and 0.725~0.775. The dotted line corresponds to the $\chi$ of
0.0 (blue), 0.81 (light red), and 1.0 (dark red), respectively.






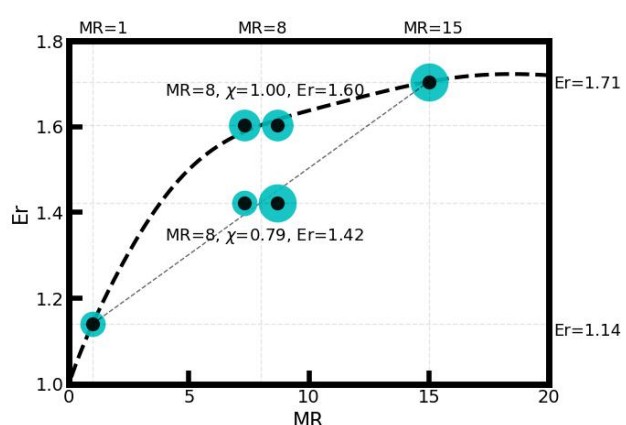

**Figure 6.** Schematic diagram that denotes the relationship between χ and Er.



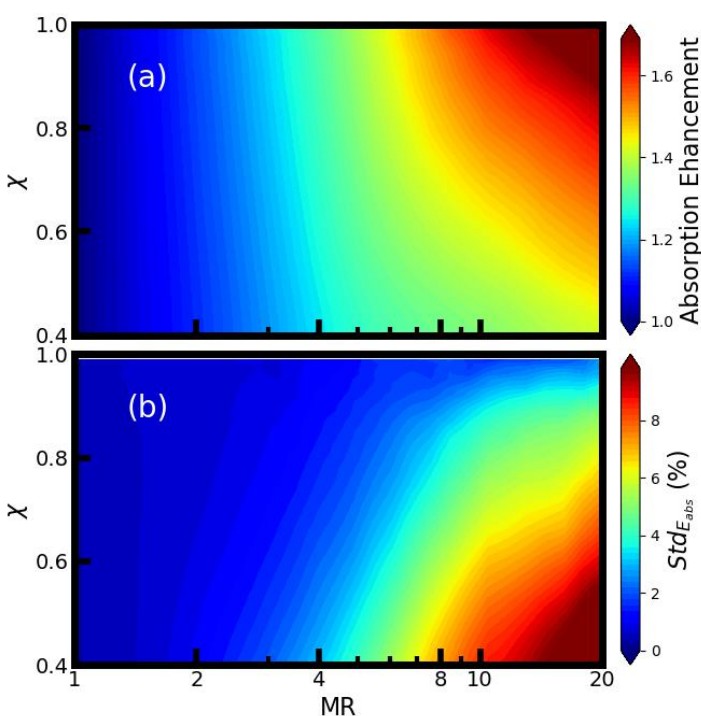


**Figure 7.** The calcaulted (a) mean $E_{abs}$ values and (b) standard deviations of the $E_{abs}$ values for
different MR and χ.






| ID | $(D_\alpha, D_\gamma)$ | $\chi$ | P1[*1] | P2[*1] | P3[*1] | P4[*1] | P5[*1] | P6[*1] | Tot[*1] |
|---|---|---|---|---|---|---|---|---|---|
| 1 | (1.00, 1.00) | - | (0, 1) | (0, 1) | (0, 1) | (0, 1) | (0, 1) | (0, 1) | (0, 6) |
| 2 | (1.00,1.56) | 0 | (1,0) | (0, 1) | (0, 1) | (0, 1) | (0, 1) | (0, 1) | (1, 5) |
| 3 | (1.00, 1.89) | 0 | (1,0) | (1,0) | (0, 1) | (0, 1) | (0, 1) | (0, 1) | (2,4) |
| 4 | (1.00, 2.00) | 0 | (1,0) | (1,0) | (1,0) | (0, 1) | (0, 1) | (0, 1) | (3,3) |
| 5 | (1.26, 2.00) | 0.26 | (2,0) | (2,0) | (0,2) | (0,2) | (1,1) | (1,1) | (6, 6) |
| 6 | (1.83, 2.00) | 0.83 | (1,3) | (1,3) | (3,1) | (3,1) | (2,2) | (2,2) | (12,12) |
| 7 | (2.00, 2.00) | 1.00 | (1,1) | (1,1) | (1,1) | (1,1) | (1,1) | (1,1) | (6,6) |
| 8 | (1.5, 1.50) | 1.00 | (1,6.1) | (1,6.1) | (1,6.1) | (1,6.1) | (1,6.1) | (1,6.1) | (6, 36.6) |
| 9 | (1.35, 1.50) | 0.70 | (1,0) | (1,0) | (1,6.1) | (1,6.1) | (1,12.2) | (1,12.2) | (6, 36.6) |

**Table 1.** Detail information of the BC particles shown in Fig.2

[*1] Mass of the BC component of and non-BC component (arbitrary unit).