# Peer review of "Method to Quantify the Black Carbon Aerosol Light Absorption Enhancement with Mixing"

_Atmospheric Chemistry and Physics, 2021_

## Author Comment (AC1)

Reviewer #1,

Thanks for your comments! The point-by-point responses are listed below.

*Comment:* In this paper, the authors proposed a method to quantify light absorption enhancement for black carbon (BC) aerosols by considering entropy and diversity. The authors indicated that the mass ratio (MR) of non-BC coating thickness to BC (MR) and particle-to-particle heterogeneity represent two key parameters in regulating the radiative absorption enhancements by BC. They introduced a BC mixing state index ( $\chi$ ) to quantify the dispersion of BC mixing states based on a binary system of BC and non-BC components.

They showed that the BC light absorption enhancement increases with  $\chi$  for the same MR, indicating that  $\chi$  can be employed as a factor to constrain the light absorption enhancement of ambient BC. This work proposed a novel framework to treat BC light absorption enhancement, which can be useful to study BC radiative effects in climate models. The paper was reasonably written, but some effort is still necessary to improve its readability. I recommend the publication of this paper in ACP, provided that the following issues have been adequately addressed.

*Reply:* Thanks for the helpful comments.

**Comment: Major points**

(1) The title needs to be modified since it is unclear how their framework is obviously linked to "entropy and diversity measures"

*Reply:* Thanks for the comment. We modified the title into "Method to Quantify the Black Carbon Aerosol Light Absorption Enhancement with Mixing State Index"

*Comment:* (2) The authors argued that the BC light absorption enhancement is dominantly determined by two physical parameters, i.e., MR and  $\chi$ . However, there are several studies showing that the chemical properties of the coating materials, i.e., organic versus inorganic species, are also critical in regulating the morphology and optical properties. For example, coating of sulfuric acid has been shown to be more efficient in altering the BC morphology and light absorption (e.g., Variability in morphology, hygroscopic and optical properties of soot aerosols during internal mixing in the atmosphere, Proc. Natl. Acad. Sci. USA 105, 10291, 2008). Such an aspect needs to be discussed in the context of their proposed framework.

*Reply:* Thanks for the helpful suggestions. The reviewer gave a good aspect that should be concerned when dealing with the light absorption of the ambient BC-containing aerosols. We added some corresponding descriptions into the introduction part.

*Comment:* (3) It would be desirable that their proposed framework can also compared to other experimental studies, particularly those relevant to different chemical species (Enhanced light absorption and scattering by carbon soot aerosols internally mixed with sulfuric acid, J. Phys. Chem. 113, 1066, 2009; Effects of dicarboxylic acid coating on the optical properties of soot, Phys. Chem. Chem. Phys. 11, 7865, 2009).

**Reply:** Thanks for the helpful comment. The reviewer gave a perspective aspect that we should concern in our future works. Our study aims to propose a method to quantify the black carbon aerosol light absorption enhancement with mixing state index, which requires the information of particle resolved BC mass ratios. Unfortunately, these data were not measured in the refered experitemental studies.

*Comment:* (4) Also, I believe that their proposed framework deals exclusively with dry particles. Under atmospheric conditions, aerosols (particularly for those containing high level of inorganic species) likely experience hygroscopic growth at high relative humidity (RH), which inevitably impacts their morphology and optical properties. How such an issue could be addressed by their method.

**Reply:** Thanks for the comment. We calculated the BC absorption coefficient under different RH conditions with Mie theory and found that the BC light absorption enhancement is slightly related to the aerosol hygroscopic growth. The core diameter and coating thicknesses were 100 nm and 50 nm with a refractive index of the core and coating of 1.67 + 0.67i and 1.46 + 0i respectively. As for the  $\kappa$  for of shell, it is estimated using the parameterization scheme proposed by Yu et al. (2018) and mean results of the real-time measurement of the aerosol chemical compositions by an In situ Gas and Aerosol Compositions Monitor (TH-GAC3000, China). A mean value of 0.16 is derived and used here. The calculated absorption coefficient ( $\sigma_{abs}$ ) under different relative humidity (RH) are shown in Fig. R1. Results show that the  $\sigma_{abs}$  ranges between 40.5 and 36.5 when the RH ranges between 40% and 90%. Thus the  $\sigma_{abs}$  varies slightly within 5% under different RH conditions. Therefore, the light absorption enhancement of BC particles at dry conditions can be employed as a good approximate value of atmospheric conditions.

**Figure R1.** Calculated  $\sigma_{abs}$  under different RH values.

*Comment:* (5) A recent work showed BC-catalyzed sulfate formation (An unexpected catalyst dominates formation and radiative forcing of regional haze, Proc. Natl. Acad. Sci. USA 117, 3960, 2020), which is primarily responsible for their optical properties under polluted conditions. How would the BC aging processes, i.e., reactive (catalyzed) versus physical (condensation/partitioning) would impact their proposed framework?

**Reply:** Thanks for the comment. The BC-catalyzed sulfate formation and condensation/partitioning process would result in the different chemical compositions of the BC coating materials, which in theory would impact the refractive index of the coating and further influence the optical properties. However, once the refractive index of the coating material is determined by other methods, the relationship between the light absorption enhancement, mass ratio, and BC mixing states can be determined.

*Comment:* Minor suggestions. Improvement in English is still necessary. I identified some grammar errors below.

Comment: Line 13, replace "its" by "the".

*Reply:* Thanks for the comment. We replace "its" with "the".

Comment: Line 14, delete "thickness".

*Reply:* Thanks for the comment. We deleted 'thickness'.

*Comment:* Line 149, add a conjunction between two parallel sentences. *Reply:* Thanks for the comment. We revised the manuscript.

Reference:

Yu, Y., Zhao, C., Kuang, Y., Tao, J., Zhao, G., Shen, C., and Xu, W.: A parameterization for the light scattering enhancement factor with aerosol chemical compositions, Atmospheric Environment, 10.1016/j.atmosenv.2018.08.016, 2018.

---

## Author Comment (AC2)

Reviewer #2,

*Comment:* This paper presents an analysis of ambient SP2 measurements at a site in Taizhou, China, to explain the range of observed black carbon absorption enhancements given a certain value of the mass ratio or non-BC coating material and BC. Motivated by the fact that previous studies show that the mass ratio and the absorption enhancement are only weakly related, the authors show that the range of absorption enhancement values at a given mass ratio can be explained by the mixing state of BC-containing particles (quantified by the mixing state metric  $\chi$ ). The paper presents an interesting analysis and fits within the scope of ACP. I have several comments that should be addressed before the paper is suitable for publication. I should note that the paper contains quite a few typos. I only flagged the typos that in my view hampered the understanding of the material, and I strongly recommend thoroughly proofread the revised version.

*Reply:* Thanks for the comments and suggestions. The point-by-point responses are listed below.

*Comment:* General comment: 1. To make the paper more impactful, I recommend that the authors could make more clear how their findings of the relationship of Eabs, MR, and  $\chi$  can be applied in practice.

**Reply:** Thanks for the comment. The new finding of our study is that the mixing state index can contribute to improvements in the accuracy of simulating the BC radiative effects. In the particle-resolved simulation of ambient aerosols, the particle-to-particle heterogeneity of BC-containing aerosols can be resolved by simply introducing the BC mixing state index  $\chi$ . Then the aerosol light absorption enhancement can be better constrained by MR and  $\chi$  and

then the radiative effects of BC can be estimated. Therefore, our framework can be employed in the model by simply introduce a BC mixing state index for better estimating the BC radiative effects.

We added these descriptions into the conclusion part of the manuscript.

*Comment:* Detailed comments: 1. Title: The title could be more descriptive of what the paper is actually about (relationship of absorption enhancement, mass ratio, and BC mixing state)

*Reply:* Thanks for the comment. We modified the title into "Method to Quantify the Black Carbon Aerosol Light Absorption Enhancement with Mixing State Index"

*Comment:*2. Abstract: Make clear that MR is used here as a bulk quantity of the population, rather than a per-particle quantity, i.e., MR here is the mass ratio of non-BC coating material in the population to BC in the population.

**Reply:** Thanks for the comment. We agree with the reviewer's opinion. We added some descriptions in the abstract to make clear the definition of MR. MR is the mass ratio of non-BC coating to BC in the population of BC-containing aerosols.

*Comment:* 3. Line 14: "coating thickness" should read "coating material" **Reply:** Thanks for the comment. We revised the "coating thickness" into "coating material".

*Comment:* 4. Line 31: should read "lensing effect" (not "effects")**Reply:** Thanks for the comment. We revised the "lensing effects" into "lensing effect".

*Comment:* 5. Line 82: Specify what is meant by "size-selected mixing states". I assume it means the distribution of BC core and non-BC coating thickness for a given total particle diameter?

**Reply:** Thanks for the comment. We agree with the reviewer's opinion and revised the "size-selected mixing states" into "size-selected distribution of BC core and non-BC coating thickness" in the corresponding text.

*Comment:* 6. Section 2.2: Add information on what size ranges the instruments can sample (and for which Eabs, MR, and  $\chi$  is determined).

**Reply:** Thanks for the comment. As noted by Zhao et al. (2020), the SP2 can only detect these BC-containing aerosols with a core diameter larger than 84 nm. The DMA select the aerosol at the range between 13.3 nm and 749.9 nm. In the following discussion, the size-resolved distribution of BC core and coating thickness are constrained in the range between 84 and 749.9 nm.

The above information was added in section 2.2 in the manuscript.

*Comment:* 7. Line 92: Notation: This should be  $\frac{d^2N}{d\log Dp \cdot d\log Dc}$  (second derivative). There are many other places in the paper where this needs to be corrected.

**Reply:** Thanks for the comment. We revised the corresponding texts in the manuscript.

Comment: 8. Line 113: "without thickness" should read "without coating"

Reply: Thanks for the comment. We replaced the "without thickness" with "without

coating".

*Comment:*9. Line 116: Notation: Dp is the total diameter, so Dp = 0 doesn't make sense. **Reply:** Thanks for the comment. It should be Dp=Dc here, and we replace the  $\sigma_{abs,Dp=0}$  with  $\sigma_{abs}(Dp = Dc)$ .

*Comment:* 10. Line 112: Notation: Given that Dp and Dc are used as independent variables, I suggest writing  $\sigma_{abs}(Dp, Dc)$  rather than putting Dp and Dc as index.

**Reply:** Thanks for the comment. We revised the text based on the reviewer's comment.

*Comment:* 11. Line 118: The use of the word "dispersion" sounds awkward. Suggest using "variability of BC mixing states" or simply "Quantifying BC mixing states".

**Reply:** Thanks for the comment. We revised the text.

*Comment:* 12. Line 139:  $H_{\alpha}$  is the average mixing entropy of the population (not of each particle).

Reply: Thanks for the comment. We revised the text.

*Comment:* 13. Section 2.4: Note that a number of different (binary) species definitions for  $\chi$  have been used in the literature, e.g. Ching et al. (2017) based their calculation on hygroscopic and non-hygroscopic species, Dickau et al. (2016) used volatile and nonvolatile species, Zheng et al. (2021) compared three different variants for  $\chi$ , one of which was based on absorbing (BC) and non-absorbing species, and Yu et al. (2020) use a metric which is very related to this paper. It would be good to cite these studies here to provide context for this paper.

Ching, J., Fast, J., West, M. and Riemer, N., 2017. Metrics to quantify the importance of mixing state for CCN activity, ACP, 17, 7445-7458

Dickau, M., Olfert, J., Stettler, M.E., Boies, A., Momenimovahed, A., Thomson, K., Smallwood, G. and Johnson, M., 2016. Methodology for quantifying the volatile mixing state of an aerosol. *Aerosol Science and Technology*, *50*(8), pp.759-772.

Yu, C., Liu, D., Broda, K., Joshi, R., Olfert, J., Sun, Y., Fu, P., Coe, H., and Allan, J. D.,

2020. Characterising mass-resolved mixing state of black carbon in Beijing using a

morphology-independent measurement method, Atmos. Chem. Phys., 20, 3645–3661,

Zheng, Z., Curtis, J.H., Yao, Y., Gasparik, J.T., Anantharaj, V.G., Zhao, L., West, M. and

Riemer, N., 2021. Estimating submicron aerosol mixing state at the global scale with machine learning and Earth system modeling. *Earth and Space Science*, 8(2), p.e2020EA001500.

**Reply:** Thanks for the comment. We added these studies in second 2.4 in the manuscript.

*Comment:* 14. Line 157: "group bulks" sounds awkward. Suggest "populations". **Reply:** Thanks for the comment. We revised the text correspondingly.

*Comment:*15. Figure 2: This figure is confusing since in line 151,  $\chi$  in this paper was defined only based on BC-containing particles (meaning that BC-free particles are not included in the calculation), while Figure 2 shows BC-free particles as examples. Please clarify and modify figure 2 as necessary.

**Reply:** Thanks for the comment. We modified figure 2. In the new figure, the amounts of BC are very small (mass ratio of the core and shell are 10-9) and most of the aerosols are composed of the non-BC component. We added the BC core in Fig. 2 and some revisions are

made in Table 1.

*Comment:* 16. Also, to make this figure easier to understand I suggest numbering the example populations according to the discussion in the text.

**Reply:** Thanks for the comment. We numbered the example populations in Figure 2.

*Comment:* 17. Figure 3:  $H_{\alpha}$  and  $H_{\gamma}$  are redundant with  $D_{\alpha}$  and  $D_{\gamma}$  but more difficult to interpret than the diversity metrics  $D_{\alpha}$  and  $D_{\gamma}$ . Suggest removing the subpanels for  $H_{\alpha}$  and  $H_{\gamma}$  from this figure.

**Reply** Thanks for the comment. We removed the subpanels for  $H_{\alpha}$  and  $H_{\gamma}$  from the Figure 3.

*Comment:* 18. Figure 3: The temporal variability of the quantities shown here is interesting and deserves more in-depth discussion. For example, in line 187, it says that " $D_{\alpha}$  decreases with the MR". However, the figure shows  $D_{\alpha}$  decreasing while MR is increasing. Please clarify and explain more clearly what process is responsible for these changes. Also, Figure 3a shows relatively low  $\sigma_{sca}$  values during the daytime while MR remains at a relatively constant level. Can you explain why this is?

**Reply:** Thanks for the comment. We tend to show that the  $D_{\gamma}$  decreases with the increasing of MR and made some revisions in the manuscript correspondingly.

The  $D_{\gamma}$  value is mainly determined by the total mass concentration ratio of the BC component to the non-BC component. It varies between 1 and 2 for different total mass concentration ratios. The  $D_{\gamma}$  increases when the mass ratio approaches 1. During the aging processing when the MR is larger than 1, it is reasonable that the  $D_{\gamma}$  is decreasing while the

MR is increasing from Fig 2. However, it is not clear why the  $D_{\alpha}$  decrease with the MR.

Also, Figure 3a shows relatively low  $\sigma_{sca}$  during the daytime while MR remains at a relatively constant level. The bulk MR values are mainly determined by aging process of Bc and new emission of BC. The aging process of BC would lead to the increment of MR, while the new emission of BC would lead to the decrement of MR. In the day time, the decrement of MR due to new emission may be comparable to that of the increment of MR due to aging process. Thus, the MR can remain at a relative constant level.

*Comment:* 19. Line 208: "refractive index of  $\chi$  --- What does this mean?

**Reply:** Thanks for the comment. We revised the text into "BC mixing states index  $\chi$ ".

*Comment:* 20. Figure 5: Suggest mentioning that the population shown for  $\chi = 0.81$  is only one possible example. There are many other possible ways the particle composition can be arranged that would give the same mixing state index.

**Reply:** Thanks for the comment. We added some descriptions in section 3.3 correspondingly.

Zhao, G., Shen, C., and Zhao, C.: Technical note: Mismeasurement of the core-shell structure of black carbon-containing ambient aerosols by SP2 measurements, Atmospheric Environment, 243, 117885, 10.1016/j.atmosenv.2020.117885, 2020.

---

## Author Comment (AC3)

**Comments:** This papers uses a combination of an SP2 with a sizing instrument to deliver information concerning BC mixing state, which is topic of much interest us. We would like to take this opportunity to make the authors aware of a previous work we have published on the subject (Yu et al., 2020), which also used monodisperse SP2 measurements to generate the Riemer-style mixing state metrics, albeit classified using a CPMA rather than a DMA. Can the authors comment on how comparable the metrics produced by the two techniques are?

**Reply:** Thanks for the comment. Yu et al. (2020) provided the detailed single-particle level mixing state information, which can contribute to future studies concerning BC lifetime and transportation to help to constrain the simulation of BC radiative effects. The mixing state metrics proposed in our method are basically the same as that of Yu et al. (2020). The derived mixing state index from the CPMA-SP2 systems without any assumptions while that from the DMA-SP2 systems with the assuming that the effective density of the BC-containing coating material are the same as that of non-BC particles.

However, our manuscript mainly focus on the relationship between the mixing state index with light absorption enhancement and demonstrating that the mixing state index can be further used in model to quantify the BC light absorption enhancement. Our study offer new insights that the mixing state index can contribute to improvements in the accuracy of simulating the BC radiative effects.

We added some discussions in the manuscript.

Yu, C., Liu, D., Broda, K., Joshi, R., Olfert, J., Sun, Y., Fu, P., Coe, H., and Allan, J. D.:

Characterising mass-resolved mixing state of black carbon in Beijing using a morphology-independent measurement method, Atmospheric Chemistry and Physics, 20, 3645-3661, 10.5194/acp-20-3645-2020, 2020.

---

## Author Response (AR2)

Thanks for your comments! The point-by-point responses are listed below.

*Comment:* The manuscript has been largely improved since the previous few versions. It emphasizes an important issue when converting the bulk MR to the absorbing properties of individual particles. The authors claim that by introducing the parameterization of the mixing state index, the variation of Eabs at the same MR could be explained. I recommend publication after a few issues addressed.

*Reply:* Thanks for the comments.

*Comment:* (1) I think it is necessary to show a few plots in the main text to demonstrate how you have measured the mobility size-resolved BC core size (Dc) distribution, e.g. a few Dp-Dc matrix for the cases in Fig. 3.

*Reply:* Thanks for the comment. The measured mean value of BC-containing number size distributions under different Dp and Dc between the day of 27 and 28, May, 28 and 29, May, 29 and 30, May are shown in Fig.R1. It is obvious that the BC-containing number and coating thickness increase with the pollution levels.

We added the Dp-Dc matrix for the cases of Fig. 3 in the manuscript and the supplementary materials.

[Figure]

**Figure R1.** The measured BC-containing aerosols under different Dp and Dc conditions during the period of (a) 27, May and 28, May, (b) 28, May and 29, May, and (c) 29, May, and 30, May. The panels (d), (e), and (f) are the corresponding BC core number size distributions of (a), (b), and (c), respectively.

*Comment:* (2) One thing still not quite clear is how you have converted the mixing state index to the absorption in bulk.

*Reply:* When the ambient aerosol $\chi$ and MR were measured, the corresponding $E_{abs}$ can be estimated from Fig. 7(a) in the manuscript.

The main purpose of our manuscript is better to constrain the difference between the measured and calculated BC-containing aerosol $E_{abs}$. We added some descriptions to the manuscript.

*Comment:* A higher Chi means the coatings were more homogenously distributed on the rBC, rather than the population with lower Chi containing a fraction of BC without apparent Eabs.

*Reply:* Thanks for the comment.

*Comment:* Section 3.4 is a bit too simplified for readers to obtain the necessary information.

*Reply:* Thanks for the comment. We added some descriptions in section 3.4 to make clear the Mont-Carlo simulations.

During the simulation, a group of the BC-containing aerosols was generated with the Dp and Dc meet the following conditions and the number of BC-containing particles was assumed to be 30. For each of the BC-containing particles, the core diameter of the BC particle was randomly generated with a geometric mean diameter of 130.7 nm and a geometric standard deviation of 1.5, which is the mean measurement results of the BC core distribution during the field measurement (Zhao et al., 2020). The corresponding MR of the BC particle is assumed to be randomly distributed in the range between 0.0 (pure BC particles without coating)

and 78.0 (particles with a core diameter of 130 nm and a total diameter of 560 nm). For each group particle, the corresponding aerosol bulk MR, $E_{abs}$ and $\chi$ can be calculated using the core-shell Mie scattering model. The simulations were conducted for $10^7$ times, and the calculated mean and standard deviation of $E_{abs}$ under different MR and $\chi$ are summarized for further analysis.

**Comment:** Line 241 says you used a constant rBC core size distribution, but Fig. 2 gives a few examples of different chi, which contain apparently different rBC core size distributions.

*Reply:* Thanks for the comment. The main purpose of the diagram in Fig. 2 is to illustrate the relationship between $D_\alpha$, $D_\gamma$, and $\chi$. For better display the condition with $D_\alpha$, $D_\gamma$, and $\chi$ values equaling 1, 1, and 1 respectively, we need to display the BC particles in Fig. 2 with different core distributions. However, it is not related to the rBC core size distributions in Line 241.

**Comment:** I would suggest giving more details about how the absorption has been calculated, to explain it is the Chi but not the rBC core size distribution causing the variation of resultant Eabs.

*Reply:* The details of calculating the single particle and bulk absorption are shown in section 2.3.1 and 2.3.2 in the manuscript. Some more detailed

descriptions were added in the manuscript.

We calculated the measured mean BC core distributions under different pollution conditions corresponding to Fig. R1 and the results are shown in Fig. R2. From Fig. R2, the normalized BC-core distributions under different pollutions are almost the same for different pollution levels as shown in Fig. R2. Thus, it is the $\chi$ that mainly causes the variation of resultant Eabs.

[Figure]

**Figure R2. Normalized BC PNSD under different pollution conditions corresponding to fig. S7. a (red), b (green), and c (blue).**

*Comment:* (3) Based on comment (1), it would be useful to see how the ambient chi varies and what is the essential reason Chi has caused different Eabs. Is low Chi because there was a large fraction of uncoated/less coated large rBC?

***Reply:*** Thanks for the comment. The reviewer gives a very perspective view about the research we are going to carry out in our future work as it is very important to see how the ambient $\chi$ varies with the ambient conditions. The $\chi$ is low partially because there was a large fraction of uncoated/less coated large rBC. From the definition of the $\chi$, the $\chi$ is low due to the high particle-to-particle heterogeneity. In our future work, the characteristic of the $\chi$ due to ambient processing such as BC emission, aging, and boundary development.

***Comment:*** (4) Also, it may not be appropriate for all to use the core-shell model when BC was thinly coated, has this been considered.

***Reply:*** Thanks for the comment. It is always overestimated when the core-shell model is used to calculating the ambient BC-containing light absorptions. The overestimation was accounted for using the correction coefficient suggested by Wu et al. (2018).

***Comment:*** (5) Has the particle shape been considered regarding the measurement of DMA, given the electrical mobility sizing is sensitive to the particle shape [Hu et al. 2021]. Particles especially at larger Dm may require consideration for the particle non-sphericity, while using mobility size may overestimate the total particle mass. This discussion may be included.

*Reply:* Thanks for the comment. We agree with the reviewer's opinion that the particle shape should be considered when calculating the BC light absorption. It is always overestimated when the core-shell model is used to calculating the ambient BC-containing light absorptions. The overestimation was accounted for using the correction coefficient suggested by Wu et al. (2018).

*Comment:* (6) Fig. 3 only shows a very narrow range of chi, would be possible to show a longer time series and cover the range of ambient measured chi.

*Reply:* Thanks for the comment. We add the time series of the measured $\chi$ time series in Fig. S6 (d) in the supplementary material. The $\chi$ ranges between 0.6 and 0.83. For a better understanding of the characteristics of the above parameters, we only present the time series of these parameters during a pollution period between 27, May and 30, May in Fig. 3 in the manuscript.

*Comment:* Other technical comments:

The font size of last paragraph should be consistent with the others. There are many places in the texts having inconsistent format, such as font size and line space.

***Reply:*** Thanks for the comment. We checked the format of the manuscript again.

***Comment:*** Line 164, typo "ss".

***Reply:*** We have deleted the 'ss' at line 164.

***Comment:*** There are grammatical errors in line 278 and line 281 "by simply introducing", line 279, duplicated "then". Please carefully check through the whole texts.

***Reply:*** Thanks for the comments. We have checked the grammatical errors of our texts.

References

Hu, K., et al, Measurements of the Diversity of Shape and Mixing State for Ambient Black Carbon Particles: Geophysical Research Letters, 48 (17), 10.1029/2021GL094522,2021.

Wu, Y., Cheng, T., Liu, D., Allan, J. D., Zheng, L., and Chen, H.: Light Absorption Enhancement of Black Carbon Aerosol Constrained by Particle Morphology, Environ Sci Technol, 52, 6912-6919, 10.1021/acs.est.8b00636, 2018.

Zhao, G., Shen, C., and Zhao, C.: Technical note: Mismeasurement of the core-shell structure of black carbon-containing ambient aerosols by SP2

measurements, Atmospheric Environment, 243, 117885, 10.1016/j.atmosenv.2020.117885, 2020.